# Long-Term Use of Oral Hygiene Products Containing Stannous and Fluoride Ions: Effect on Viable Salivary Bacteria

**DOI:** 10.3390/antibiotics10050481

**Published:** 2021-04-22

**Authors:** Anne Brigitte Kruse, Nadine Schlueter, Viktoria Konstanze Kortmann, Cornelia Frese, Annette Anderson, Annette Wittmer, Elmar Hellwig, Kirstin Vach, Ali Al-Ahmad

**Affiliations:** 1Department of Operative Dentistry & Periodontology, Faculty of Medicine, University of Freiburg, 79106 Freiburg, Germany; viktoria.konstanze.kortmann@gmail.com (V.K.K.); annette.anderson@uniklinik-freiburg.de (A.A.); elmar.hellwig@uniklinik-freiburg.de (E.H.); ali.al-ahmad@uniklinik-freiburg.de (A.A.-A.); 2Division for Cariology, Department of Operative Dentistry and Periodontology, Faculty of Medicine, University of Freiburg, 79106 Freiburg, Germany; nadine.schlueter@uniklinik-freiburg.de; 3Clinic for Oral, Dental and Maxillofacial Diseases, Department of Conservative Dentistry, University Hospital Heidelberg, 69120 Heidelberg, Germany; cornelia.frese@med.uni-heidelberg.de; 4Institute of Medical Microbiology and Hygiene, Department of Microbiology and Hygiene, Faculty of Medicine, University of Freiburg, 79104 Freiburg, Germany; annette.wittmer@uniklinik-freiburg.de; 5Institute of Medical Biometry and Statistics, Faculty of Medicine, University of Freiburg, 79104 Freiburg, Germany; kv@imbi.uni-freiburg.de

**Keywords:** stannous ion, fluoride, salivary bacteria, culture technique, viable bacteria

## Abstract

The aim of this randomized, controlled clinical trial was to isolate and identify viable microorganisms in the saliva of study participants that continuously used a stannous and fluoride ion (F/Sn)-containing toothpaste and mouth rinse over a period of three years in comparison to a control group that used stannous ion free preparations (noF/Sn) over the same time period. Each group (F/Sn and noF/Sn) included 16 participants that used the respective oral hygiene products over a 36-month period. Stimulated saliva samples were collected at baseline (T0) and after 36 months (T1) from all participants for microbiological examination. The microbial composition of the samples was analyzed using culture technique, matrix-assisted laser desorption ionization–time of flight (MALDI–TOF) mass spectrometry, and 16S rDNA *Polymerase Chain Reaction* (PCR). There were only minor differences between both groups when comparing the absolute values of viable microbiota and bacterial composition. The treatment with F/Sn led to a slight decrease in disease-associated and a slight increase in health-associated bacteria. It was shown that the use of stannous ions had no negative effects on physiological oral microbiota even after prolonged use. In fact, a stabilizing effect of the oral hygiene products containing stannous ions on the health-associated oral microbiota could be expected.

## 1. Introduction

For several decades, various fluoride compounds such as sodium fluoride or amine fluoride have been successfully used in preventative dental care. It is well known that fluoride ions have remineralizing properties and therefore a cariostatic effect, which is more or less independent of the type of fluoride compound used. Furthermore, a bacteriostatic effect is also attributed to the fluoride ions if used in higher concentrations [1], as various glycolysis-related enzymes, predominantly enolase, in the bacterial metabolism, can potentially be inhibited by fluoride ions. In addition to fluoride compounds containing monovalent counter ions, compounds containing polyvalent metal cations such as stannous ions are also available for use. Stannous fluoride (SnF_2_) was one of the first fluoride compounds used in toothpastes in the United States, with the initial clinical studies on its use published in the 1950s [2]. The divalent stannous ions can, on the one hand, be retained on and incorporated into enamel and dentin [3], resulting in reduced hard tissue solubility [4]. On the other hand, stannous ions are bacteriostatic and exhibit bactericidal activity by inhibiting further bacterial enzyme activity in the glycolysis of bacteria. Accordingly, stannous ions and fluoride ions have complementary properties and can thus prevent plaque formation. Supragingival dental hard tissues exhibit colonization of microbiota immediately after cleaning by brushing or professional tooth cleaning by planktonic microbial species from saliva. Oral biofilms develop rapidly through the adherence of primary colonizers to the tooth surface covered by the saliva pellicle followed by building complex communities with a variety of interrelationships between different microbiota. During plaque formation, different stages have been described, during which the complexity of interdependent microorganisms embedded in an organic polymer matrix increases [5]. 

Another mode of action that has been discussed is that stannous ions, which are incorporated into the dental hard tissue, reduce the number of potential calcium binding sites on the hydroxyapatite, thus inhibiting the accumulation of bacteria on the tooth surface via calcium bridges [6]. For this reason, stannous ions in combination with fluoride are widely used in modern dentistry, both for the inhibition of demineralization of the dental hard tissue and for the prevention of bacterial-driven inflammatory processes. 

In general, metal cations have a good substantivity in the oral cavity and show plaque-inhibiting effects for up to six hours [2,7]. Studies that investigated the effect of stannous ions on bacterial colonization in the oral cavity found that in combination with fluoride, stannous ions could inhibit the vitality of the biofilm and reduce the total bacterial count in the oral cavity and on the mucosa [8]. Furthermore, SnF_2_ toothpaste was shown to be more effective in reducing salivary bacterial counts in vivo when compared to a sodium fluoride (NaF) formulation over a period of 5 days [8]. In an in vitro study using live-/dead-staining, these findings were confirmed in a three-species biofilm model using SnF_2_ containing toothpaste [9]. Here, a reduction of the amount of extracellular polymeric substance (EPS) was found. However, in a clinical study with children who were treated overnight with a 0.4% SnF_2_ gel for 3 weeks, no significant effect could be found on the bacterial plaque ecology [10]. This could in part be due to the short period of application. 

The major drawback of most of the studies to date is that the study duration was frequently short, and therefore, no conclusion could be drawn from the results regarding the implications of long-term use of the oral hygiene products evaluated. One can assume that in the oral cavity of a healthy individual, balanced physiological microbiota can be found that are considered to be commensal [10]. Unfavorable environmental changes such as poor oral hygiene, high levels of sugar consumption or the use of anti-microbial agents, however, might shift this ecology towards potentially pathogenic species. This imbalance can contribute to the development of oral diseases associated with bacteria, such as caries and/or gingivitis [11,12]. 

The present study on microbiota in saliva is part of a larger clinical study on the long-term effects of stannous and fluoride ion-containing oral hygiene products on various oral health parameters, including salivary parameters, caries, gingivitis and erosive tooth wear, in healthy endurance athletes [13]. A preceding analysis focused on the salivary microbiome using 16S rDNA high-throughput sequencing. A beneficial effect for the test group using stannous and fluoride-containing oral hygiene products for a 36-month period compared to a control group using only stannous ion-free oral hygiene products was determined. The study showed persistency of bacterial species that are associated with oral health, including members of the genera *Neisseria* and *Granulicatella*, in the test group [14]. Since 16S rDNA high-throughput sequencing is not able to differentiate between viable and non-viable bacteria, the impact of long-term use of stannous and fluoride ion containing oral hygiene products on detectable viable species is still pending. In this context, the culture technique is the approach of choice. The aim of this study was to identify viable microorganisms in saliva samples from healthy volunteers who used oral hygiene products containing stannous and fluoride(F/Sn) ions over a 36-month period in comparison to a control group that used oral hygiene products containing sodium fluoride but no stannous ions (noF/Sn) over the same period.

## 2. Materials and Methods

### 2.1. Study Design

This study was part of a larger randomized controlled clinical trial [13] that followed the guidelines of Good Clinical Practice and conformed to the Helsinki declaration. Ethical approval was obtained from the local ethics committee (original proposal S-566/2012, amendment S-230/2016; University of Heidelberg). The study was registered at the German Clinical Trials Registry Platform (DRKS00005019, registration 2013/05/27). This report follows the criteria of the CONSORT statement [15]. All enrolled participants signed a data privacy statement and an informed consent form.

### 2.2. Participants

Saliva samples from 32 endurance athletes who were recruited at the University of Heidelberg, Germany were examined. Criteria for inclusion in the study included at least 5 h of endurance training per week and overall good health. Exclusion criteria for participation included pregnancy or lactation, the intake of antibiotics within the last 30 days prior to the commencement of the study or being either a student or staff member of the Faculty of Dentistry. For further information on recruitment, inclusion and exclusion criteria and study drop-out see Appendix A and [13]. After randomization, the subjects were allocated to either the test or the control group.

### 2.3. Intervention

Participants of the test group were supplied with stannous and fluoride ion-containing dentifrice and mouth rinse, which were used by them on a daily basis. The mouth rinse contained 800 ppm stannous ions (Sn^2+^ from SnCl_2_), 500 ppm fluoride (375 ppm F^-^ from NaF and 125 ppm F^-^ from amine fluoride (AmF)) (Elmex Erosion Protection Mouthrinse, CP GABA, Hamburg, Germany). The toothpaste contained 3500 ppm stannous ions (Sn^2+^ from SnCl_2_), 1400 ppm fluoride (700 ppm F^-^ from NaF and 700 ppm F^-^ from AmF) and 0.5% chitosan (Elmex Erosion Protection Toothpaste, CP GABA, Hamburg, Germany). Participants were told to use the mouth rinse once daily for 30 s. The toothpaste had to be used twice daily for regular oral hygiene. Participants in the control group maintained their usual oral hygiene routine without any additional instructions using conventional toothpaste containing 1500 ppm fluoride (from NaF) and no stannous compound. All participants were instructed in the usage of the oral hygiene products at each appointment, which were conducted at 6-month intervals.

### 2.4. Clinical and Microbiological Examination

Clinical parameters including an erosive tooth wear score (Basic Erosive Wear Examination Index: BEWE [16]) and a caries score (International Caries Detection and Assessment System: ICDAS [17]) were collected from all participants at baseline and at 6-month intervals until the end of the study period at 36 months. The clinical parameters were collected following a professional teeth cleaning procedure at each appointment [13]. All clinical examinations were performed by one blinded and calibrated examiner at the Department of Conservative Dentistry, University Hospital Heidelberg (Heidelberg, Germany). Saliva samples were collected at two time points within the study period, at baseline and after 36 months. Microbiological examinations were performed approximately half a year after collecting the last saliva sample at T1 for both time points, baseline (T0) and after 36 months (T1). 

### 2.5. Collection and Storage of Samples

Saliva samples were taken after stimulation of saliva production by chewing on sterile paraffin. Participants were instructed not to eat, drink, brush their teeth or smoke within 1 h prior to sampling. After collection, the saliva samples were deposited in a vessel containing reduced transport medium fluid (RTF) and frozen until analysis. Basic parameters such as flow rate, pH value and buffer capacity were examined as described in [13]. The analysis of saliva buffering capacity was performed using a commercial test kit (Saliva Check Buffer, GC EUROPE, Leuven, Belgium).

### 2.6. Isolation and Identification of Bacterial Species

The frozen saliva samples (−80 °C) were thawed in a water bath at 36 °C prior to homogenization for 30 to 60 s using a Vortex (Bender und Hobein GmbH, Laboratory Medical Technology, Germany). Dilution series of the samples were plated onto two different culture media to obtain countable single colonies of the bacterial species they contained. The viable bacterial species were grown in countable single colonies to enable their identification and to distinguish between the different bacterial types. Aerobic and facultative anaerobic bacteria were cultivated on Columbia blood agar plates (CBA) at 37 °C and 5–10% CO_2_ atmosphere for 5 days (Heraeus Holding GmbH, Hanau, Germany). Anaerobic bacteria were isolated on yeast-cysteine blood agar plates (HCB) at 37 °C for 10 days in an anaerobic chamber (GENbox bioMérieux, Marcy l’Etoile, France) (Figure 1), and all colony types were sub-cultivated to obtain pure cultures. Single colonies were assessed based on their shape and color, which served as a preliminary classification of the bacterial species.

The colony forming units (CFU) were determined as follows:Bacteria number/mL = colony number × dilution grade

To identify the bacteria, pure cultures of the bacterial isolates were prepared from all chosen individual colonies. All pure bacterial isolates were then identified using MALDI–TOF MS (matrix-assisted desorption ionization time-of-flight mass spectrometry) analysis in a MALDI Biotyper Microflex LT as described in detail in previous research [16]. In brief, single pure colonies were picked up and analyzed by MALDI–TOF according to the manufacturer’s recommendations. The obtained mass spectra of each pure bacterial isolate were compared with a database containing 3740 reference spectra (representing 319 genera and 1946 species) using BioTyper 3.0 software. This comparison delivered a similarity level which was described as a log score. A score value of ≥2000 delivered identification on the species level, whereas a genus level identification was gained by score of ≥1700. A score value below 1700 indicated no significant similarity of the spectrum obtained with any database entry. If the results obtained were questionable, the procedure was repeated. Identification by means of polymerase chain reaction (PCR) and sequencing of the 16S rDNA genes was used if the bacterial isolates could not be identified by MALDI–TOF MS. 16S rDNA sequencing was performed as described earlier in detail [17] to identify the bacterial isolates on a species level. In brief, DNA extraction from each pure isolate of Gram-negative bacteria was conducted using a lysis buffer (10 mmol/L Tris-HCl buffer, 1 mmol/L ethylenediaminetetraacetic acid, 1% Triton X-100, pH 8.0) and heating at 100 °C for 10 min. The lysis buffer including the DNA was then centrifuged at 12,000 g for 10 min and 1 µL of the supernatant was used for the subsequent PCR. To extract the DNA from Gram-positive bacteria, the QIAamp DNA Mini Kit (Qiagen, Hilden, Germany) was used according to the manufacturer’s instructions. The amplification of the 16S rRNA gene was conducted in a total volume of 50 µL containing 2 U Taq Polymerase (Qiagen, Hilden, Germany), 200 mol/L each of deoxyribonucleoside triphosphate (dNTP) and 5 µL 10X PCR-buffer (Qiagen, Hilden, Germany), 300 nmol/L of reverse- and forward-primer and MgCl2 (2.5 mmol/L). The primer pair used for amplification of the 16S rRNA gene comprised a forward-primer (TP16U1: 5′-AGAGTTTGATC[C/A]TGGCTCAG-3′) and a reverse-primer (RT16U6: 5′-ATTGTAGCACGTGTGT[A/C]GCCC-3′). The 1018 base pair PCR products were extracted and purified using the GFX PCR DNA and gel band purification kit (Amersham Biosciences Europe GmbH, Freiburg, Germany), and the purified PCR products were subsequently sequenced using the BigDye terminator kit v1.1 cycle sequencing kit (Applied Biosystem, Darmstadt, Germany) and the ABI 310 Genetic Analyzer (GMI, Inc, Ramsey, MN). TP16U1 was used as a sequencing primer. The sequences obtained were analyzed using the BLAST program from the NCBI (http://www.ncbi.nih.gov/BLAST, accessed on 22 November 2020) to identify the bacterial species.

### 2.7. Blinding and Randomization

The participants were randomized by block randomization (sequentially numbered envelopes) into either the test or control group. For further information on blinding see [13].

### 2.8. Statistical Analysis

For a descriptive analysis, the mean and standard deviation were calculated. Intergroup comparisons were performed in two ways: (i) the absolute values of both groups (F/Sn, noF/Sn) were directly compared at both time points (T0, T1) and (ii) the differences between T0 and T1 (change in occurrence of bacteria) were calculated for each group prior to being compared between the groups (intergroup differences). T-tests were used for independent samples to evaluate the differences in bacterial counts between the groups (F/Sn, noF/Sn), while paired t-tests were used to analyze the differences within one group between the two time points (T0, T1). The level of significance was set to *p* < 0.05, and the statistical analysis was performed using STATA 14.2 (StataCorp LLC, College Station, TX, USA).

## 3. Results

### 3.1. Demographic Data

The demographic data of the study participants are shown in Table 1.

### 3.2. Summary of Clinical Parameters 

As the detailed clinical data have been published elsewhere [13], only the saliva related data is shown here. The salivary flow at baseline (T0) and at the end of the study period (T1) was 1.7 ± 0.8 mL/min and 2.0 ± 0.7 mL/min in the control group and 2.0 ± 0.7 mL/min and 2.33 ± 1.0 mL/min in the test group, respectively. The mean pH value at baseline (T0) was 6.7 ± 0.4 for the control and 6.8 ± 0.3 for the test group and slightly changed to 7.0 ± 0.4 and 7.0 ± 0.7, respectively, over the study period. Overall, no significant differences for these parameters could be found, neither between the groups at baseline and after 36 months nor for changes over the study period within one group. None of the participants had carious lesions with dentin involvement according to the ICDAS system.

### 3.3. Microbiological Data

Data are given as means. Standard deviations are found in Appendix A.

#### 3.3.1. Total Bacterial Counts

Microorganisms could be isolated from the saliva samples of all participants. Both at baseline (T0) and after 36 months (T1), no significant difference was found between the two groups; furthermore, no significant difference between both groups regarding the change in bacteria between T0 and T1 (intergroup differences) was found (*p* = 0.1679). Over the study period from baseline to 36 months, the total bacterial counts in salivary samples from the test group slightly increased (not significant) from 7.10 to 7.45 log10 CFU/mL, while the control group showed a significant increase from 7.16 to 7.88 log10 CFU/mL (*p* < 0.001). 

#### 3.3.2. Microbial Composition

In the samples from the 32 participants of both groups, a total of 80 different microbial species could be detected. No significant difference between the two groups was found at baseline (T0; *p* = 0.8773) and after 36 months (T1; *p* = 0.0652). In addition, no significant difference in terms of a change of diversity between T0 and T1 (intergroup differences) was found (*p* = 0.1207) between the two groups. The mean number of different species increased significantly both within the test group and the control group from baseline to the end of the study period (test group: baseline 10.75, end 13.00, *p* = 0.013; control group: baseline 10.88, end 15.00, *p* < 0.001). The bacterial composition of viable species consisted of facultative anaerobic Gram-positive cocci, facultative anaerobic Gram-positive rods, aerobic Gram-negative cocci, facultative anaerobic Gram-negative rods, strictly anaerobic Gram-negative rods, anaerobic curved Gram-negative rods, strictly anaerobic Gram-negative cocci and Gram-positive anaerobic and Gram-negative rods and cocci (Figure 2). The detected anaerobic Gram-negative rods in the control group comprised *Prevotella denticola*, *P. intermedia*, *P. nigrescens*, *P. histicola*, *P. melaninogenica*, *P. tannerae*, *P. jejuni*, *P. salivae*, *P. pallens* and *P. nanceinsis*. In the test group, *P. tannerae*, *P. histicola*, *P. nigrescens*, *P. melaninogenica*, *P. jejuni*, *P. salivae* and *P. pallens* were detected.

Overall, the most common bacterial species identified in the sampled saliva were *Streptococcus oralis*, *S. mitis*, *S. parasanguinis*, *S. salivarius*, *Granulicatella adiacens*, *Actinomyces odontolyticus*, *Rothia mucilaginosa*, *R. dentocariosa*, *R. aeria*, *Neisseria subflava*, *N. flavescens*, *N. perflava*, *Veillonella parvula*, *V. dispar* and *Atopobium parvulum*. For detailed data on the distribution and amount of individual bacterial species, see Appendix A.

Between the groups, significant differences at baseline (T0) as well as after 36 months (T1) were only found for *S. anginosus* (T0 *p* = 0.0034; T1 *p* < 0.001) and *Gemella/Granulicatella* spp. (T0 and T1 *p* < 0.001). Additionally, at T1, fewer *Streptococcus* spp. were detected in the test group compared to the control group (*p* = 0.0439). For all other species, no significant differences were found, neither at the baseline (T0) nor after 36 months (T1). Intergroup differences measured via the difference between T0 and T1 were found for Gram-negative bacteria (*p* = 0.042), black-pigmented bacteria (*p* = 0.016) and Gram-negative anaerobic rods (*p* = 0.033), which showed a significantly higher prevalence in the control group (Figure 2 and Figure 3).

Within one group, from baseline to the end of the study, significantly higher counts of several species and subgroups of species were found in the control group (Figure 2, Figure 3 and Figure 4). These included all Gram-positive bacteria (*p* < 0.001), aerobic species (*p* < 0.001), anaerobic species (*p* = 0.006), Gram-positive aerobic cocci (*p* = 0.003), Gram-negative aerobic cocci (*p* < 0.001), black-pigmented bacteria (*p* < 0.001), *Actinomyces* species (*p* = 0.003), *Streptococcus* species (*p* = 0.003), *S. mitis* (*p* = 0.004) and *Veillonella* species (*p* < 0.001). No significant differences regarding the aforementioned bacteria were found for the test group. Both groups showed significantly higher levels at the end of the study period compared to baseline for all Gram-negative species (test *p* = 0.001, control *p* < 0.001), Gram-positive aerobic rods (test *p* < 0.001, control *p* = 0.016), Gram-negative anaerobic rods (test *p* = 0.005, control *p* < 0.001), *Rothia* species (test *p* = 0.006, control *p* = 0.001) and *Neisseria* species (test and control *p* < 0.001).

## 4. Discussion

In the present study, the 16 participants in the test group used a toothpaste and mouth rinse containing stannous and fluoride ions (F/Sn) over a study period of 36 months, while the 16 participants in the control group used oral hygiene products with no stannous ions but containing fluoride (noF/Sn). To our knowledge, this is the first study that investigates the effect of the long–term use of F/Sn oral hygiene products by evaluating the survival and dynamic of bacterial species in salivary samples using the culture technique. 

In general, only minor differences were found between both groups. In the case of a comparison of the absolute values between groups at both time points, the differences found in *S. anginosus* (higher abundance in the test group) and *Gemella/Granulicatella* spp. (lower abundance in the test group) were detectable both at baseline and after 36 months. These differences, therefore, cannot be related to the use of stannous ion-containing oral hygiene products. The values for all *Streptococcus* species were lower in the test group compared to the control group after the long-term use of stannous ions. However, the differences between both groups were relatively small, and the clinical relevance in this context was questionable, as for the single subspecies of streptococci no difference was found between the two groups. This clearly showed the stability of the commensal oral microbiota and that stannous ions had no negative impact on the bacterial composition over a long period.

Intergroup differences, measured via the difference between T0 and T1, were found for Gram-negative bacteria, black-pigmented bacteria and Gram-negative anaerobic rods, with an increase in the control group compared to the test group. This confirms the findings of the Illumina Miseq Sequencing in another investigation of this study series [14]. Some *Prevotella* spp. belong to the group of black pigmented bacteria and are associated with periodontal disease and various other oral diseases, including tumors [18,19]. More *Prevotella* species associated to the orange complex of periodontitis bacteria [20] such as *P. denticola*, *P. intermedia*, *P. nigrescens*, *P. histicola*, *P. melaninogenica* and *P. tannerae* were found in the control group. To the contrary, almost half of the detected *Prevotella* species such as *P. jejuni*, *P. salivae* and *P. pallens* in the test group did not belong to the pathogenic group of subgingival bacteria [20]. The increase in total bacterial counts in the test group was not significant. Hence, such an increase could be explained by the natural dynamic of the salivary bacterial load. Additionally, salivary bacteria exist in a semi-planktonic state that may cause a natural variation in their numbers in human saliva. Among others, *P. histicola* was isolated most frequently. It was first identified in the human oral cavity in 2008 and is reported to be associated with a high prevalence of caries and also with the pathogenesis of early childhood caries [21,22]. Consequently, one can say that the lower abundance in the test group can be associated with a more health-associated microbiota. However, the higher abundance of black pigmented species in the control group might also be the result of a higher percentage of males in this group (94% compared to 75% in the test group). Zaura et al. [23], in their investigation of the salivary microbiome in a large study, found that certain *Prevotella* species occur significantly more often in males than in females. 

As toothpastes with a fluoride concentration of a maximum of 1450 ppm fluoride were used by the participants in the present study, most likely, only fluoride concentrations in the saliva and the dental plaque were achieved, which could have an impact on solubility of the dental hard tissue but not on the bacterial metabolisms [1]. Therefore, the effect of the oral hygiene products on the bacteria found in the present study can predominantly be attributed to the effect of the stannous ions. There is evidence from in vitro studies that the growth of Gram-negative anaerobic taxa such as *Fusobacterium nucleatum* and *Porphyromonas gingivalis* might be suppressed by stannous ions [9,24], which was confirmed by the present study. In a short-term clinical study over 4 days, the positive effect of stannous fluoride in plaque reduction was reported [25]. This effect could also be shown in other in vivo studies [8,26]. Furthermore, a clinical study over 6 months showed a reduction in gingival inflammation compared to the control group after using an amine fluoride/stannous fluoride (AmF/SnF_2_) toothpaste [27]. Therefore, not only the short-term use of stannous ion containing preparations shows beneficial effects, but also the long-term use of stannous ions in combination with fluoride seems not to negatively affect the microbial composition. One can even speculate that stannous fluoride leads to the maintenance of a stable oral microbiota and suppresses the increase of some adverse anaerobic bacterial species, supporting a healthy homeostasis of oral bacterial population.

During the study period of 36 months, some culturable bacteria considerably increased, but only in the control group. These included Gram-positive microbiota such as *Actinomyces* spp., which belong to the commensal microbiota of human saliva [28]. They are often found particularly in the sulcus and proximal areas and are able to ferment glucose to succinic, acetic and lactic acids [29]. As they are acid-tolerant, they are primarily involved in the development of caries such as *Actinomyces viscosus* and *A. naeslundii* in the case of caries on root surfaces [30,31]. *A. naeslundii* is also considered to be a biofilm producer and is associated not only with caries but also with gingivitis [29]. The amount of *S. salivarius* and the strictly anaerobic Gram-negative *Veillonella* species increased from T0 to T1 in the control group. *S. salivarius* is considered to be the primary colonizer, and subsequently the primary binding site, for *Veillonella* species; for example. *S. salivarius* is potentially acidogenic [32], while *Veillonella* spp. are most commonly found in dental plaque and are often associated with bacteria that are able to ferment carbohydrates to lactic acid such as strains of *Actinomyces, Streptococcus* and *Lactobacillus* [33]. *Lactobacillus* spp. are also known to play a role in the development of caries by producing acids and contributing to demineralization [34]. These results taken together would indicate an increase in cariogenicity; however, the results of the caries scores taken at each appointment were in clear contrast to these findings. It is most likely that one reason for the increase was as a result of the long-term storage of the saliva samples, which is discussed in further detail in the section dealing with the limitations of the study. It is worth noting that in the test groups this increase was not measurable. For this reason, the increase in the control group should not simply be regarded as an increase in the caries risk, but rather, the lack of increase in the test group could possibly be viewed as a reduction in caries risk by the introduction of the stannous ions, or more generally spoken, as a stabilization of the oral microbiota. This hypothesis is supported by the significant increase in diversity in the test group, which appears to be related to oral health [35]. In contrast to these findings, the Illumina Miseq sequencing showed no differences for either alpha-diversity or beta-diversity between the test and control groups [14]. It should be emphasized that up to 50% of oral bacteria cannot be isolated using the culture technique, as was shown using the culture-independent 16S-rDNA cloning technique [36]. A comprehensive comparison of culture analysis with the culture-independent cloning technique to investigate the endodontic microbiota revealed the limits of determination of the oral bacterial diversity by isolation of the bacteria on agar plates [37]. Additionally, the building of flocs may cause bias during determination of the CFUs from salivary bacteria as has been shown by a comparison of culture technique, DAPI-staining and fluorescence in situ hybridization [38]. However, biases using the culture-independent molecular methods, which could be caused by the different levels of efficiency of DNA extraction methods, PCR amplification, sequencing technique or the post-run bioinformatic analysis pipeline, should also be considered [39,40]. Moreover, free DNA from dead bacterial cells is also detected using modern, culture-independent, next-generation sequencing methods [40], which could lead to the over-estimation of specific findings. Therefore, to adequately assess the effects of the long-term use of active agents such as stannous ions in combination with fluoride, it is important to investigate viable and active oral bacteria. It should be kept in mind that oral health is a result of a balance between the oral microbiota and the host [41].

One challenge arising from the analysis of biological samples from two widely separated time points is that different storage times can affect the results. Therefore, the data of the study were analyzed in three ways to minimize the risk of bias in interpretation of the results. The highest risk of bias may derive from the long-term storage of samples from T0, as this could lead to lower bacterial growth within the samples [42]. This could be an explanation for the increase of total bacterial load as well as the increase in diversity for both groups when comparing the values of T0 with those of T1. It should be emphasized that a modification of the microbiota would result independently of the storage medium after such a long-term period of storage. This should be taken into account when interpreting the results presented here and therefore, the results of the direct comparison between T0 and T1 should be interpreted with caution. Changes observed in the control group should be interpreted more as the normal conditions, while changes or lack of changes in the test group, as found in the present study, should be interpreted as an effect. When looking at differences between the groups, however, storage effects can be expected for both groups. Therefore, the detected intergroup differences may be interpreted more confidently, as they are most likely to mainly be related to the use of the stannous ions in oral hygiene products. To include both in the analysis of the intergroup differences, i.e., the absolute effect of stannous ions and the effect of the stannous ions over the whole study period, the comparison between the groups was performed in two different ways, namely by the direct comparison of the values of both groups at each time point and by the comparison of the changes between both time points of both groups. The challenge of storage time still affected the latter results; however, we could better show differences based on individual changes. The direct comparison of the absolute values at one time point excludes a potential storage bias, but it does not provide information on changes due to study participation. Such effects of sample storage should be avoided in future studies by culturing the bacteria a short time after taking the samples from the participants.

In addition to the study limitations mentioned, a further limitation could be attributed to the fact that the participants included in the study belong to a special population, i.e., endurance athletes. This study group was chosen because the intention of this study was to investigate the effect of stannous ion-containing oral hygiene products on various oral parameters inclusively on the occurrence and the development of erosive tooth wear. For this purpose, patients at high risk for erosive tooth wear were needed. Endurance athletes are at particular risk for this dental disease, as they eat a special diet rich in acids and often consume large amounts of sports drinks [43]. They represent, therefore, a relatively homogenous group of participants at risk for exogenously caused erosive tooth wear. Often, persons at high risk for erosive tooth wear show no biofilm-related diseases in the oral cavity such as carious-caused demineralization and periodontal diseases, which could possibly also influence the microbiological results of the saliva collected. Therefore, the results of the study have to be estimated against this background. Comparable long-term studies should thus be conducted with patients at higher risk for biofilm-associated intraoral diseases to find larger differences between groups. 

## 5. Conclusions

This in vivo study showed that the long-term use of the tested oral hygiene products containing stannous ions did not negatively affect the commensal oral microbiota of the participants. After 36 months, the microbial community of the test group consisted of a health-associated microbiota with a lower concentration of potentially pathogenic anaerobic Gram-negative taxa than found in the control group at the same time point. In conclusion, oral hygiene products containing stannous ions in combination with fluoride can be recommended for long-term use without adverse effects on the homeostasis of the healthy microbiota.

## Figures and Tables

**Figure 1 antibiotics-10-00481-f001:**
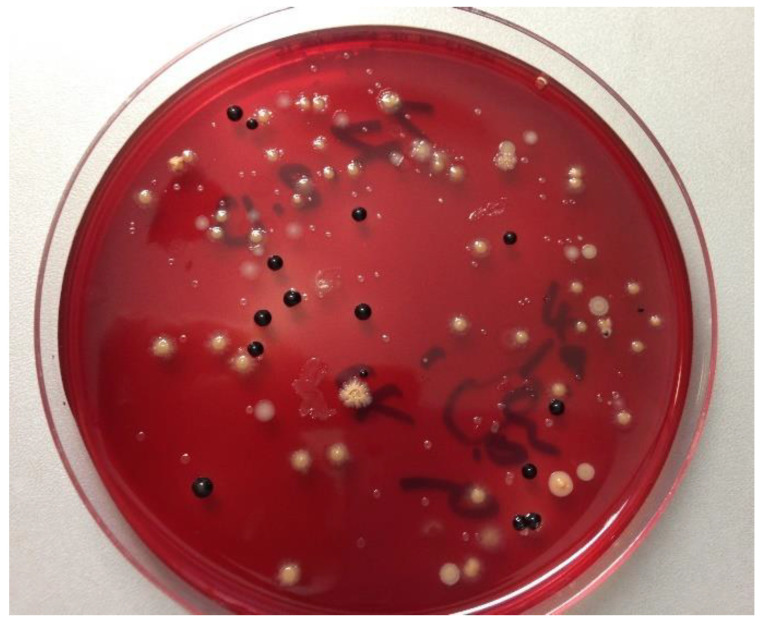
Anaerobic bacteria in culture. Different bacterial colonies which were anaerobically cultured from total salivary bacteria on an HCB agar plate for 10 days.

**Figure 2 antibiotics-10-00481-f002:**
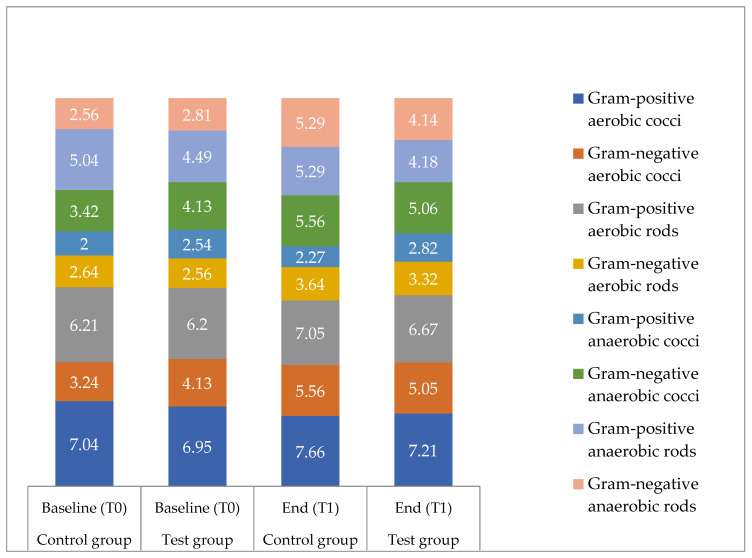
Subgroups of bacterial species detected in saliva. Bacterial counts of subgroups of bacterial species for the test group and the control group at baseline (T0) and after 36 months (T1); values are given as means in log10 CFU/mL; *n* = 16 for each group.

**Figure 3 antibiotics-10-00481-f003:**
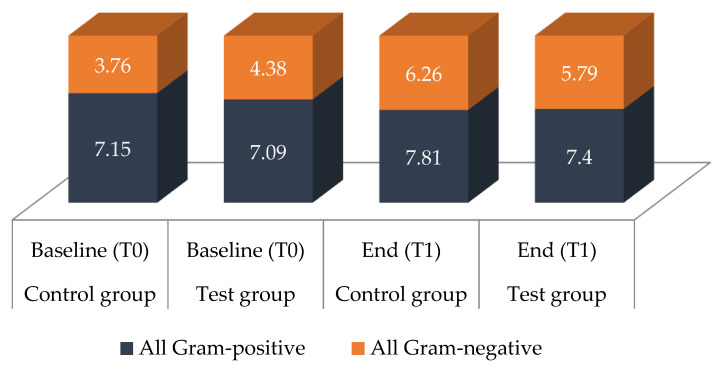
Gram-negative and Gram-positive species detected in saliva. Bacterial counts of Gram-negative and Gram-positive bacterial species for the test group and the control group at baseline (T0) and after 36 months (T1); values are given as means in log10 CFU/mL; *n* = 16 for each group.

**Figure 4 antibiotics-10-00481-f004:**
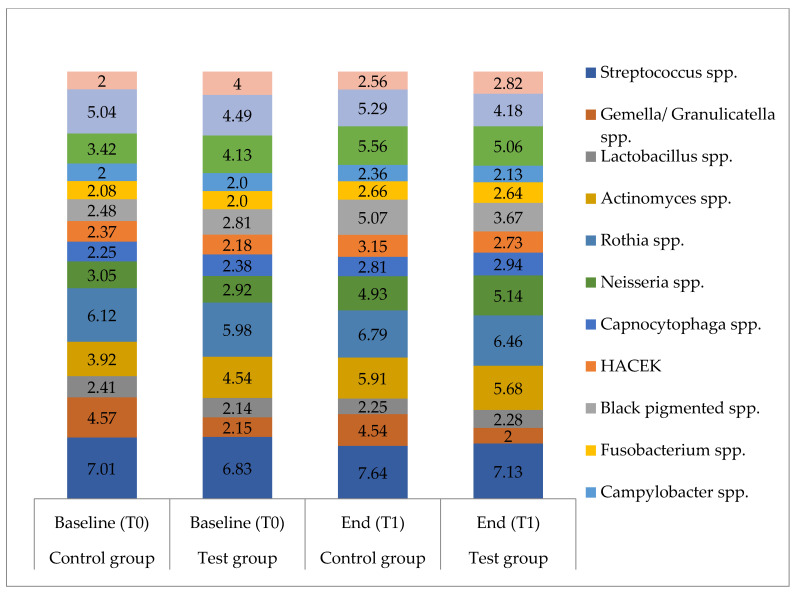
Microbial composition of saliva samples in Log_10_ CFU/mL. Bacterial counts of bacterial species for the test group and the control group at baseline (T0) and after 36 months (T1); values are given as means in log_10_ CFU/mL; spp. = species pluralis, HACEK = *Haemophilus* spp., *Aggregatibacter actinomycetemcomitans*, *Aggregatibacter aphrophilus*, *Cardiobacterium hominis*, *Eikenella corrodens*, *Kingella* spp.; *n* = 16 for each group.

**Table 1 antibiotics-10-00481-t001:** Demographic data.

Participants (*n* = 32)	Test Group (*n* = 16)	Control Group (*n* = 16)
Age	35.8 ± 10.8	34.9 ± 8.52
Gender	12 (75%) male 4 (25%) female	15 (94%) male 1 (6%) female

The values are given as means with standard deviations.

## Data Availability

The data presented in this study are available on request from the corresponding author.

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
