# Peer review of "Long-Term Use of Oral Hygiene Products Containing Stannous and Fluoride Ions: Effect on Viable Salivary Bacteria"

_antibiotics, 2021, doi:10.3390/antibiotics10050481_

Round 1

Reviewer 1 Report

Comment 1: Ln. no. 28 and 29, give abbreviations for MALDI-TOF and PCR, respectively.

Comment 2: Ln. 196, please write the information about statistics below the table.

Comment 3: In writing results, no need to give information about standard deviations, better write in a separate table.

Comment 4: When you write the first time give the full scientific name for the genus and species. When you write subsequently, no need to write full genus name... just for example S. mitis enough. Please check throughout the manuscript about scientific names.

Comment 5: In figures, no need to write the figure title inside. In figure legends, you already mentioned.

Comment 6: In figures just write n=16, why separately writing, if it is different ok. otherwise no need.

Comment 7: If possible include bacterial species figures in agar plates or microscopic one. Comment 8: Rewrite discussion and conclusion.

Author Response

Comment 1: Ln. no. 28 and 29, give abbreviations for MALDI-TOF and PCR, respectively.

Answer to the reviewer: Thank you for this useful comment. We changed it in the text. (see p.1,line 28-30)

Comment 2: Ln. 196, please write the information about statistics below the table.

Answer to the reviewer: We changed this in the text. (see p.6,line 236)

Comment 3: In writing results, no need to give information about standard deviations, better write in a separate table.

Answer to the reviewer: As suggested, we gave only means in the text. Standard deviations are found in supplement 2 now. (see page 6, line 247, 254-255, 262-263)

Comment 4: When you write the first time give the full scientific name for the genus and species. When you write subsequently, no need to write full genus name... just for example S. mitis enough. Please check throughout the manuscript about scientific names.

Answer to the reviewer: Thank you for this comment. We changed it in the text. (see p.7,line 267-275,279,290;p.10, line 326,348;p.11, line 377,379)

Comment 5: In figures, no need to write the figure title inside. In figure legends, you already mentioned.

Answer to the reviewer: As suggested, the titles were removed from the figures. (see figures 2-4)

Comment 6: In figures just write n=16, why separately writing, if it is different ok. otherwise no need.

Answer to the reviewer: We changed this in the text below the figures. (see p.8, line 298-301; p.9, line 304-306; p.9, line 308-309)

Comment 7: If possible include bacterial species figures in agar plates or microscopic one.

Answer to the reviewer:A figure of different bacterial colonies which were anaerobically cultured was integrated into the text as Figure 1.

Comment 8: Rewrite discussion and conclusion.

Answer to the reviewer: We optimized the discussion and conclusion and added supporting references where appropriate. (see p. 9, line 320, p.10, line 338-342, 353-357; p.11, line 368,407-408; p. 12, line 430-432, 435-446,455-456)

Reviewer 2 Report

This study being part of a larger study compared cultivable microbiota in saliva in individuals using in the test group a stannous and fluoride containing toothpaste and mouthrinse over three years. The major parts of the study (clinical data, 16S rDNA sequencing) have been published somewhere else, this manuscript deals with cultivable microorganisms. The study is well written, standard laboratory methods were used. However, as several limitations were raised by the authors themselves it is unclear (at least for me) how much this study increases our knowledge in oral microbiota.  

Introduction

Please discuss the bacteriostatic properties of fluoride related to the achievable concentration in vivo.

Please shortly introduce biofilm formation on teeth as the readers of journal are not mainly dentists.

Methods: Why did the study focus on endurance athletes only? For how long saliva samples have been stored until analysis?

How did the authors check compliance of the study participants?

Results: How did the authors explain the increase of total counts in the test group? For me, the mean number of cultivable different species is quite low and the authors found only rarely anaerobically growing gram-negative rods. In more detail the identified gram-negative rods should be described and the microbiological results should be related to oral health in the study population.

Please exchange the term flora by “microbiota”.

Author Response

This study being part of a larger study compared cultivable microbiota in saliva in individuals using in the test group a stannous and fluoride containing toothpaste and mouthrinse over three years. The major parts of the study (clinical data, 16S rDNA sequencing) have been published somewhere else, this manuscript deals with cultivable microorganisms. The study is well written, standard laboratory methods were used. However, as several limitations were raised by the authors themselves it is unclear (at least for me) how much this study increases our knowledge in oral microbiota.  

Introduction:

Comment 1: Please discuss the bacteriostatic properties of fluoride related to the achievable concentration in vivo.

Answer to the reviewer: We have added a short discussion on this issue (see p.10, line 353-357)

Comment 2: Please shortly introduce biofilm formation on teeth as the readers of journal are not mainly dentists.

Answer to the reviewer: We added a short introduction of biofilm formation in the text. (see p.2, line 53-59)

Methods:

Comment 3: Why did the study focus on endurance athletes only?

Answer to the reviewer: The intention of the study was to investigate the effect of stannous ion containing oral hygiene products on various parameter. The primary aim of the study was the assessment of the impact on development and progression of erosive tooth wear in persons under high risk for this disease. Endurance athletes are at high risk for developing dental erosion, as they show a special diet and often consume high amounts of sports drinks. Therefore, only this group was included. For more clarity, the respective paragraph in the discussion dealing with the choice of study groups was modified (see p 12, lines 435-446).

Comment 4: For how long saliva samples have been stored until analysis?

Answer to the reviewer: The baseline samples have been stored for approximately three and a half years, the samples of the second collection time point for approximately half a year, depending on inclusion date at baseline. This information has been added to the M&M section (see p 4 line 137-139).

Comment 4: How did the authors check compliance of the study participants?

Answer to the reviewer: The participants of the test group were supplied with the stannous containing oral hygiene products at each appointment (every six months) and were instructed in usage of them. The control group did not receive specific products, but were instructed to use conventional sodium fluoride containing, stannous ion free products. Respective information was added to the M&M section (see p 3 lines 117-118 and 126-128).

Results:

Comment 5: How did the authors explain the increase of total counts in the test group? For me, the mean number of cultivable different species is quite low and the authors found only rarely anaerobically growing gram-negative rods. In more detail the identified gram-negative rods should be described and the microbiological results should be related to oral health in the study population.

Answer to the reviewer:

Regarding the increase of total counts in the test group, the following text has now been added to the discussion: “The increase of total bacterial counts in the test group was not significant. Hence, such increase could be explained by the natural dynamic of salivary bacterial load. Additionally, salivary bacteria exist in a semi-planktonic state which may cause a natural variation of its number in human saliva.”  (see p.10, line 342-345)

As for the second point regarding anaerobic Gram-negative roads the following text has now been added to the results: “The detected anaerobic Gram-negative roads in the control group comprised Prevotella denticola, Prevotella intermedia, Prevotella nigrescens, Prevotella histicola, Prevotella melaninogenica, Prevotella tannerae, Prevotella jejuni, Prevotellla salivae, Prevotella pallens and Prevotella nanceinsis. In the test group, P. tannerae, P. histicola, P. nigrescens, P. melaninogenica, P. jejuni, P. salivae and P. pallens were detected.” (see p. 7, line 267-271)

And the following text has now been added to the discussion: More Prevotella species associated to the orange complex of periodontitis bacteria such as P. denticola, P. intermedia, P. nigrescens, P. histicola, P. melaninogenica, P. tannerae were found in the control group. In contrary, almost half of detected Prevotella species such as P. jejuni, P. salivae and P. pallens in the test group don’t belong to the pathogenic group of subgingival bacteria (Socransky et al. 1998).

(see p.10, line 338-342)

Comment 5: Please exchange the term flora by “microbiota”.

Answer to the reviewer: thank you for this helpful comment. We changed the term throughout the text. (see p.1, line 34;p.10, line 331; p. 11, line 371,390,397,408; p.12, line 451)

Round 2

Reviewer 2 Report

The authors addressed well my raised concerns.